# Nanoscratch Testing of 3Al$_2$O$_3$·2SiO$_2$ EBCs: Assessment of Induced Damage and Estimation of Adhesion Strength

Carlos Alberto Botero [1,2,*], Laura Cabezas [1,3], Vinod Kumar Sarin [4], Luis Llanes [1,3] and Emilio Jiménez-Piqué [1,3]

1    CIEFMA, Department of Materials Science and Engineering, Campus Diagonal Besòs-EEBE, Universitat Politècnica de Catalunya, 08019 Barcelona, Spain
2    Department of Quality Technology and Mechanical Engineering, Sports Tech Research Centre, Mid Sweden University, 83125 Östersund, Sweden
3    Barcelona Research Centre in Multiscale Science and Engineering, Campus Diagonal Besòs-EEBE, Universitat Politècnica de Catalunya, 08019 Barcelona, Spain
4    Department of Mechanical Engineering, Boston University, 110 Cummington Street, Boston, MA 02215, USA
*    Correspondence: carlos.botero@miun.se or sportstechresearch@miun.se

**Abstract:** In this study, the structural integrity of mullite (3Al$_2$O$_3$·2SiO$_2$) films, deposited on silicon carbide (SiC) substrates using chemical vapor deposition (CVD), was investigated via increasing load nanoscratch tests. The films were configured by mullite columns of stoichiometric composition growing from a silica-rich layer in contact with the SiC substrate. Controlled damage was induced in the 3Al$_2$O$_3$·2SiO$_2$ films at relatively low scratch loads. Radial and lateral cracking were applied until final delamination and repeated chipping were achieved as the load increased. The intrinsic integrity of the 3Al$_2$O$_3$·2SiO$_2$ film and the performance of the coated 3Al$_2$O$_3$·2SiO$_2$/SiC system, regarded as a structural unit, were analyzed. With the aid of advanced characterization techniques at the surface and subsurface levels, the configuration and morphology of the damage induced in the coated system by the nanoscratch tests were characterized, and the scratch damage micromechanisms were identified. Finally, the adhesion of the film, in terms of energy of adhesion and interfacial fracture toughness, was determined using different models proposed in the literature. The results from this investigation contribute to the understanding of the mechanical performance and structural integrity of EBC/SiC-based systems, which over the past few years have increasingly been implemented in novel applications for gas turbines and aircraft engines.

**Keywords:** nanoscratch; thin films; mullite; energy of adhesion; interfacial fracture toughness

## 1. Introduction

Coatings and thin films are often implemented to help achieve the specific properties necessary to meet the functional requirements imposed by certain applications [1]. Although such target properties are not always structural or mechanical, investigating the mechanical behavior of coatings and the structural integrity of coating/substrate systems is critical for practical implementation. One good example of this is the case of environmental barrier coatings (EBCs) based on mullite (3Al$_2$O$_3$·2SiO$_2$).

As a consequence of more than two decades of focused research and development, it has been demonstrated that EBCs are effective tools to protect Si-based substrates—especially SiC/SiC Ceramic Matrix Composites (CMCs)—from the severe pitting corrosion and recession typical of gas turbine applications [2–10]. Today, EBC-coated SiC/SiC CMCs are used in high pressure shrouds in 7F gas turbines as well as hot components of the LEAP aircraft engine [7,8]. One of the most critical challenges to tackle is reliability and durability, since failure of EBCs reduces the lifespan of CMC components, with thermal and thermo-mechanical strains shown to be one of the key contributors to EBC degradation [8,11–13]. Therefore, it is of key importance to investigate the structural integrity and durability of

EBC/CMC systems. Reports on the mechanical performance and structural integrity of such systems can be found in the literature [14–16], and interfacial adhesion and toughness issues have recently been addressed [17–19].

Within the context of protective coatings for Si-based materials, an effective approach has been to deposit mullite coatings by chemical vapor deposition (CVD) on SiC substrates [5,20–22]. This process yields coatings with a columnar microstructure, in which grains of mullite nucleate and form a thin silica-rich layer; the composition of the emerging coating approaches the value of stoichiometric mullite, i.e., when Al/Si $\approx$ 3. The columns start to grow once nucleation takes place, and their composition can be tailored to incorporate a broad range of increasing Al/Si ratios for optimum corrosion protection [5].

With regard to the configuration of the coatings described above, it is crucial from a structural standpoint to investigate their mechanical integrity. This is particularly true for the initial microns of coating, where $3Al_2O_3 \cdot 2SiO_2$ columns of stoichiometric composition grow from the thin layer of the interface with the SiC substrate. In this regard, nanoindentation and nanoscratch are shown to be the most suitable and widely used techniques to assess the mechanical properties and integrity of small volumes of materials, such as small-sized second phases and thin films [23–27].

In a previous contribution by the authors [16], the mechanical behavior of $3Al_2O_3 \cdot 2SiO_2$ films about 1 µm in thickness was investigated by means of nanoindentation using different tip geometries. The main mechanical properties of the films were assessed, and an initial analysis of their structural integrity was attempted by performing indentations with a cube corner tip. Increasing the indentation load in discrete steps was found to result in controlled cracking of the film that progressed until delamination. The film fracture toughness ($K_f$), as well as values for the adhesion energy ($G_{int}$) and interfacial fracture toughness ($K_{int}$) of this system, was determined.

In this work, the structural integrity of the $3Al_2O_3 \cdot 2SiO_2$ films and thin-coated system $3Al_2O_3 \cdot 2SiO_2$/SiC, intended as a structural unit (i.e., accounting for the film, interface, and substrate influences), is characterized using increasing load nanoscratch tests conducted on the polished surface of the films by means of a Berkovich indenter. These nanoscratches allow for controlled damage to the films that progresses at higher loads to promote film delamination. The configuration and morphology of the damage, as well as how it changes with increasing load, are evaluated and analyzed. A new range of values for film adhesion in terms of adhesion energy $G_{int}$ and interfacial fracture toughness $K_{int}$ is then provided on the basis of the induced delamination.

## 2. Materials and Methods

### 2.1. Materials and Sample Preparation

Thin $3Al_2O_3 \cdot 2SiO_2$ films of stoichiometric composition were deposited on SiC substrates. Deposition was carried out in a hot-wall CVD reactor using the system of gases $AlCl_3$–$SiCl_3$–$CO_2$–$H_2$. Details of the experimental procedures are described in the references [6].

For the mechanical testing, three coated samples were manually polished down to 3 µm diamond paste and then prepared with colloidal silica. The film thickness after polishing was $t = 1.07 \pm 0.15$ µm (as measured from FIB cross sections in at least 10 different sample areas) and the surface roughness $R_a < 6$ nm, as measured by means of AFM. A cross section of one of the films, obtained from a FIB-cut, is presented in the micrograph in Figure 1. The $3Al_2O_3 \cdot 2SiO_2$ film can be observed as a bright gray layer deposited on top of the dark gray silicon carbide substrate. As can be discerned from the figure, the film microstructure is made up of a nanolayer at the interface between the substrate and film, beneath the columnar and crystalline $3Al_2O_3 \cdot 2SiO_2$ grains.

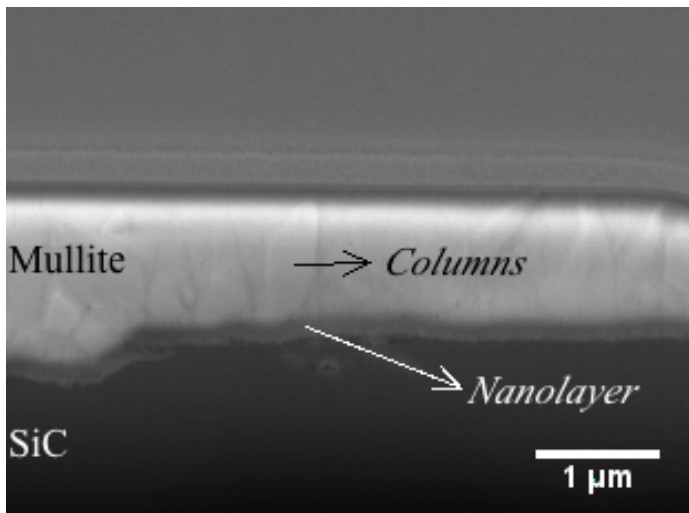

**Figure 1.** FIB cross section of a polished film. The columnar microstructure and nanolayer are indicated.

### 2.2. Nanoscratch Tests

Nanoscratch tests were performed using an XP nanoindenter (MTS, Oak Ridge, TN, USA). A Berkovich indenter was used to perform increasing load nanoscratches on the polished surfaces of the studied films at a continuous scratch rate of 1 μm/s. Tests were conducted to evaluate the intrinsic structural integrity of the mullite films without significant substrate influence, as well as the coated system mullite/SiC. Two load conditions were set for the experiments: (i) maximum load of $P_{max}$ = 50 mN and scratch length of $l_{max}$ = 50 μm (loading rate of 1 mN/s; referred to as the "low-load" condition) and (ii) maximum load of $P_{max}$ = 500 mN and scratch length of $l_{max}$ = 200 μm (loading rate of 2.5 mN/s; referred to as the "high-load" condition).

The low- and high-load conditions allowed for controlled damage to the film to promote delamination. The tip penetration during the loading (scratching) and post-loading stages of the test, as well as the friction force and coefficient of friction, were measured. Measuring these characteristics provided a simple means to determine the critical load for delamination in the coated system, as coating failure or detachment typically results in a sudden change in these parameters. This information was analyzed and contrasted with the surface and subsurface characterization of the damage induced alongside the scratch track and surrounding material.

### 2.3. Surface and Subsurface Damage Characterization

At the surface, the residual nanoscratch tracks were evaluated by means of optical microscopy (OM), laser scanning confocal microscopy (LSCM) using an Olympus LEXT OLS 3100 microscope, field emission scanning electron microscopy (FESEM) using a FESEM Neon 40 (Carl Zeiss, Oberkochen, Germany), and atomic force microscopy (AFM) using a Veeco Dimension 3100.

FIB was used to prepare cross sections in specific locations along the residual nanoscratch tracks to characterize the induced damage at the subsurface. Cross sections were done with a Zeiss Neon 40 FIB. The samples were initially pre-coated with a fine carbon layer to prevent charging locally and minimize ion beam damage. First, the scratched zones for the cross sections were located and imaged before sputtering. Subsequently, platinum layers were deposited on selected locations on the surface of the imprints to protect the surface while milling. In these areas, FIB was used to prepare trenches using a Ga⁺ ion beam with decreasing currents perpendicular to the surface. The dimensions of the trenches allowed for characterization of the damage and deformation scenario for the entire thickness of the film and a significant portion of the substrate. Finally, SEM images of the polished cross sections were acquired.

## 3. Results and Discussion

### 3.1. Low-Load Nanoscratches

Figure 2a includes the curves of penetration as a function of scratch distance, corresponding to three scratches performed at the low-load condition. It is evident from the curves that penetration increases as the indenter progresses along the scratch track both during loading and post-loading. It is also evident that the curves are continuous, with no sudden change in penetration, indicating that the integrity of the films is retained throughout the tests at the low-load condition. It can also be seen that considerable elastic recovery is displayed by the film once the load is removed after the scratch test (40% recovery at maximum load), as indicated by the differences in penetration between the scratching and post-scratching curves (see the indicative gray arrows in Figure 2a).

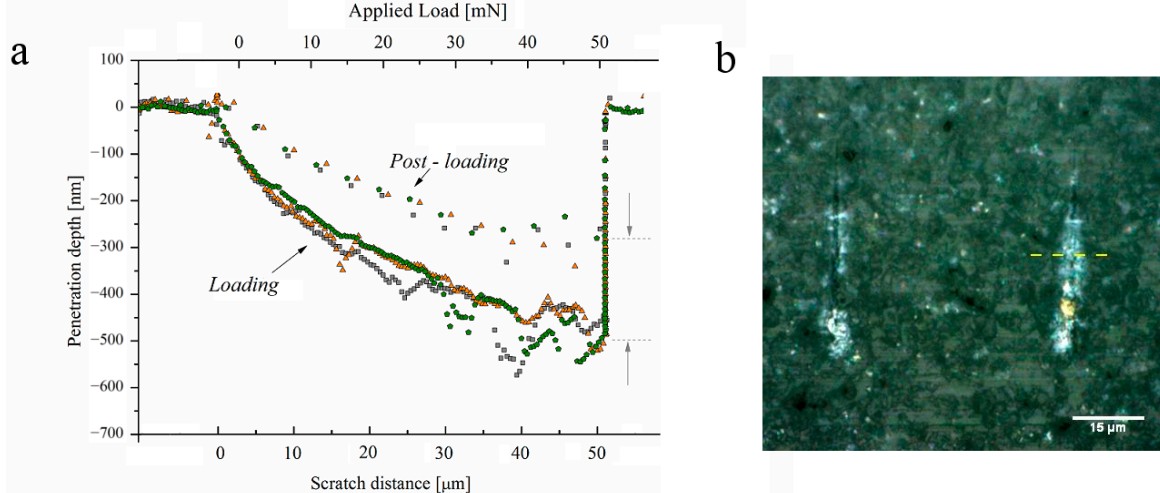

**Figure 2.** (**a**) Penetration depth as a function of scratch distance and applied load for scratches performed at the low-load condition; loading and post-loading stages are shown. (**b**) Optical micrograph of the scratches at $P_{max} = 50$ mN-$l_{max} = 50$ μm.

Two examples of residual tracks corresponding to low-load scratches are presented in Figure 2b. The optical translucence of the $3Al_2O_3 \cdot 2SiO_2$ film reveals some features of the underlaying substrate in the background of the image. There is also no evidence of failure at the surface of the film, such as cracking, spallation, or chipping, in this micrograph. This indicates that the films have good tolerance to the normal and lateral forces exerted by the indenter during the scratch experiment.

From the image in Figure 2b, it is not possible to discern whether damage was caused underneath the surface of the film. Therefore, an FIB cross section was performed near the center of one of the scratch tracks (as indicated by the dotted line of Figure 2b, at $l \approx 25$ μm and $P \approx 25$ mN) to investigate the possible damage scenario below the surface. The micrograph in Figure 3a displays a general view of this cross section; in the micrograph in Figure 3b, a central area of the scratch track is magnified.

The micrograph in Figure 3a confirms that no cracking or removal of the film material occurred around the scratch track at the surface level, as suggested by the optical micrographs. The entire coating system, including the mullite film and some portions of the SiC substrate, is visible in the cross section micrograph in Figure 3b. In this micrograph, two cracks can be identified. There is a small radial crack within the mullite film directly underneath the midpoint of the scratch track. This crack is comparable to those produced in similar films by nanoindentation using cube corner indenters [9]. It is evident from the micrograph in Figure 3b that the crack is contained in the film and does not progress further into the substrate or propagate along the interface. The second crack is located within the SiC substrate and clearly propagates laterally following an intergranular path. This substrate crack coincides with the interface on one side of the scratch track.

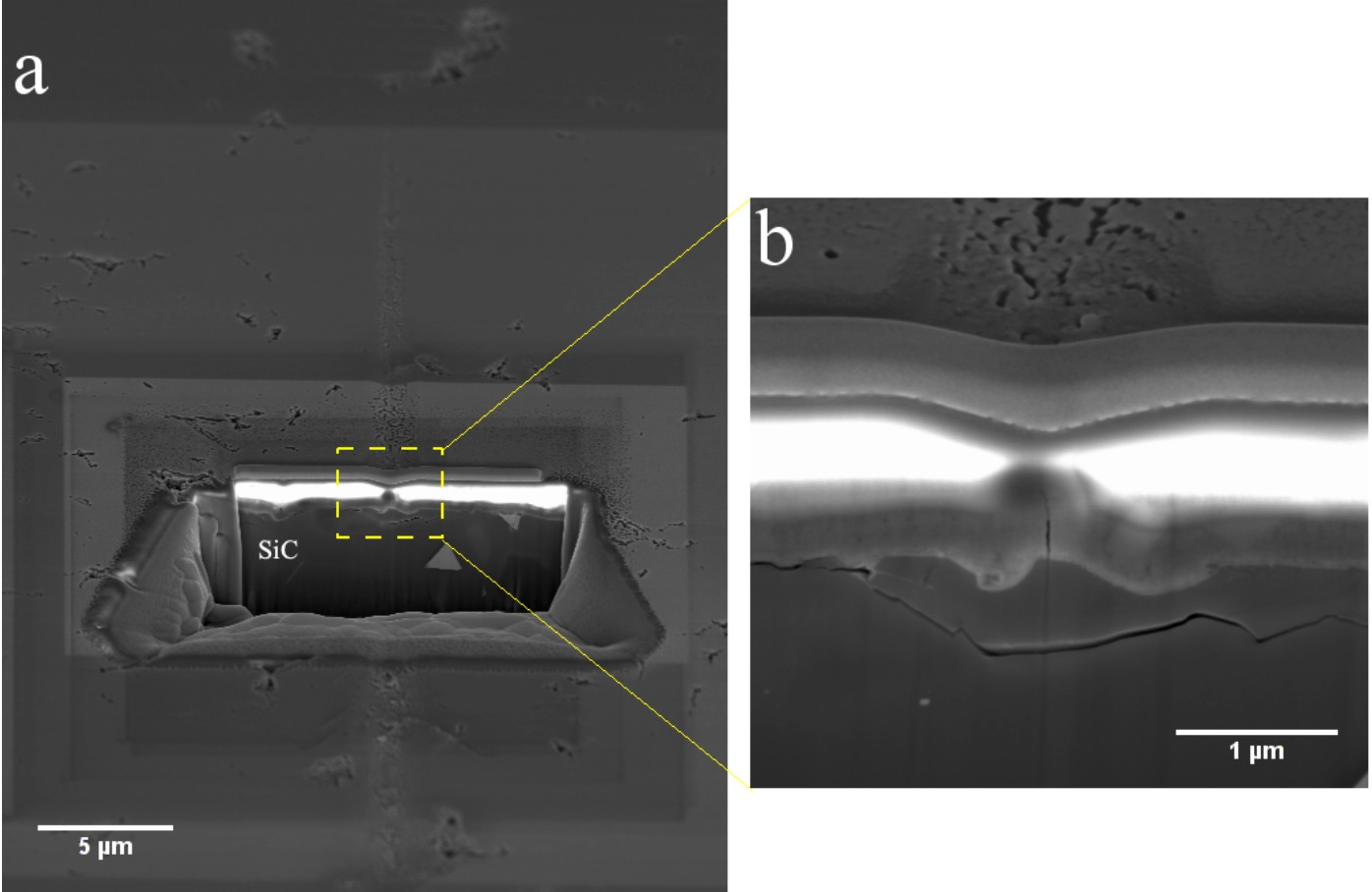

**Figure 3.** (**a**) Cross section performed using FIB at the midpoint of the scratch, i.e., at $P_{max}$ = 25 mN - $l_{max}$ = 25 μm. (**b**) Higher magnification image focused on the center of the track.

Under the low-load condition, only slight cracking damage is induced by the nano-scratch tests. Apart from the incipient radial cracking produced in the $3Al_2O_3 \cdot 2SiO_2$ film by the normal applied force, it may be that the coated system behaves as a structural unit under the low load regime since the load applied by the indenter to the film is transferred to the substrate, as evidenced by the cracking in the silicon carbide substrate.

### 3.2. High-Load Nanoscratches

In the optical image in Figure 4a, the residual track of one of the high-load condition scratches is presented. Two occurrences of damage are evident from the scratch imprint. A well-marked "bright" zone, referred to as zone 1, can be observed around the track at lower loads (as identified by the black arrow in Figure 4a). The brighter color potentially suggests a fracture event underneath the surface. As the load is raised, multiple occurrences of chipping are observed on several portions of the $3Al_2O_3 \cdot 2SiO_2$ film around the scratch track (referred to here as zone 2, as indicated by the white arrow in Figure 4a). The damage observed in this investigation is similar to that reported for micro and nanoscratch tests of comparable systems, such as nitride-based films deposited on silicon [28], TiB$_2$-based nanostructured coatings [29], sol-gel coatings on glass [30], and Al/AlN thin films deposited on Si (100) substrates [31].

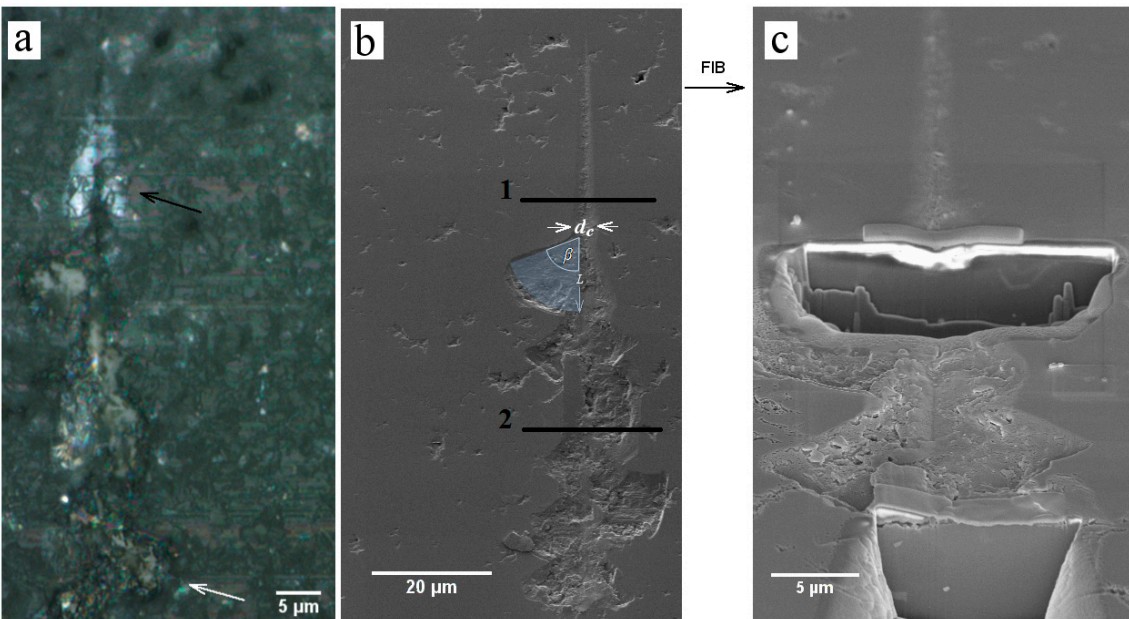

**Figure 4.** (**a**) Optical micrograph showing one of the high-load scratches; the main damage events are indicated by arrows. Surface SEM images of a high load nanoscratch (**b**) before and (**c**) after the FIB sectioning process.

The chipping on the surface of the mullite film is clearly evidenced by the SEM micrograph in Figure 4b. In this case, FIB was used to prepare two cross sections at locations along the residual scratch track to investigate the subsurface damage. As indicated by Figure 4b,c, the locations were selected based on the damage shown in the optical micrographs: bright zone (1) and chipped zones (2). Higher magnification SEM micrographs documenting the specific features of these cross sections are presented in Figures 5 and 6 respectively.

From the cross section of Figure 5 (corresponding to zone 1 of Figure 4b), three different types of cracks can be identified:

(i) A lateral crack following an intergranular path. This crack is located within the SiC substrate on one side of the scratch track, as observed in Figure 5c.

(ii) A radial crack, contained within the $3Al_2O_3 \cdot 2SiO_2$ film. This crack is oriented parallel to the loading normal axis and aligned with the center of the scratch track; it resembles that observed in the low-load scratches presented in Figure 3b.

(iii) Lateral cracks also contained within the $3Al_2O_3 \cdot 2SiO_2$ film. These cracks are parallel to the film surface and propagate sideways departing from the midpoint of the scratch. The cracks cut the film columns transversally along their propagation paths, as can be observed in Figure 5b,c.

Additionally, the nature of the chipping that occurs in the film at higher scratch loads can be evaluated from the FIB cross section images in Figure 6. The brittle character of the fracture that occurs inside the film can be observed at the surface. The residual portions of the film after delamination can also be observed at the subsurface, on either side of the track. A crack located at the interface at the left-hand side of the image suggests that the mullite film is chipped out from the SiC substrate after delamination.

The curves of penetration depth vs. applied load and distance for three of the conducted scratch tests are presented in Figure 7. An increase in the penetration is again recorded as the indenter moves along the track. The nanoindenter registers a sudden change in the penetration at a scratch length of $l \approx 40$ μm, which corresponds to a load of $P \approx 110$ mN. This discontinuity event recorded in the penetration is also observed for the friction coefficient (Figure 7b) and the friction force (Figure 7c), plotted as a function of the scratch length (and the applied load).

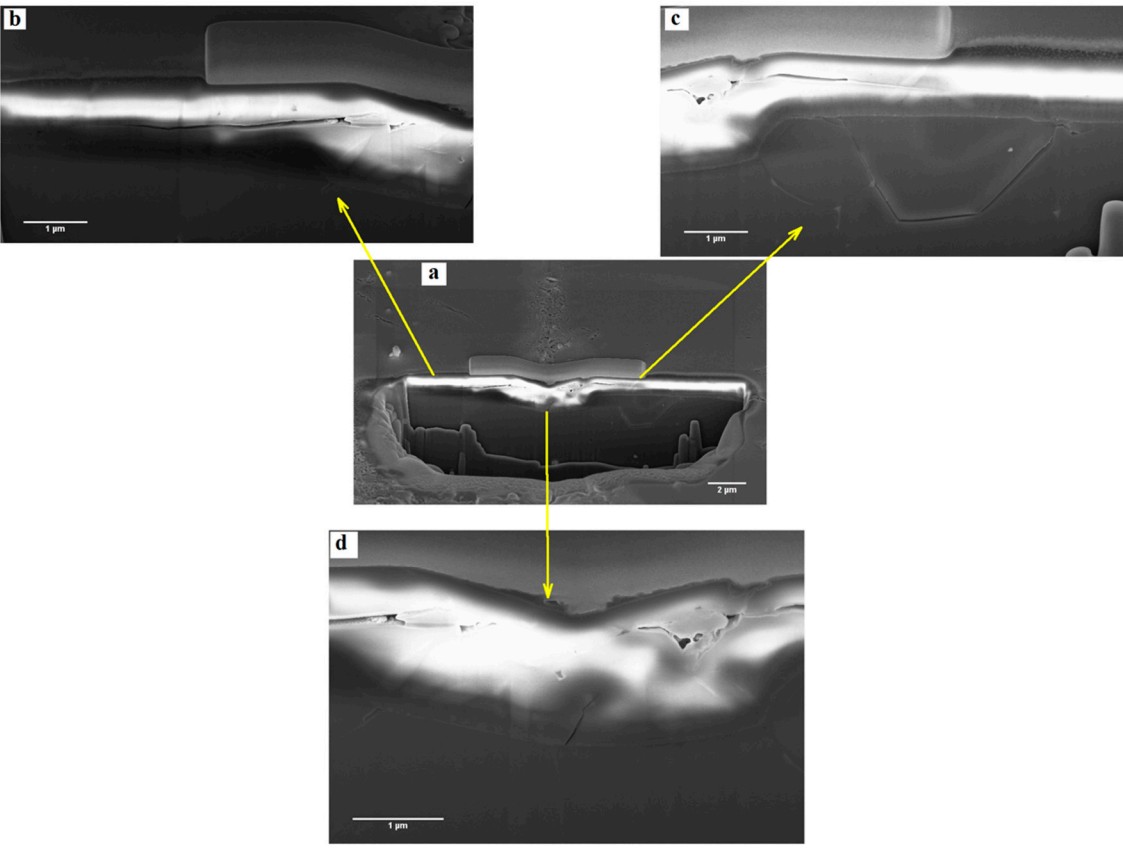

**Figure 5.** FIB cross section of zone 1 (refer to Figure 4b). General (**a**), lateral (**b**,**c**), and central (**d**) details of the cross section.

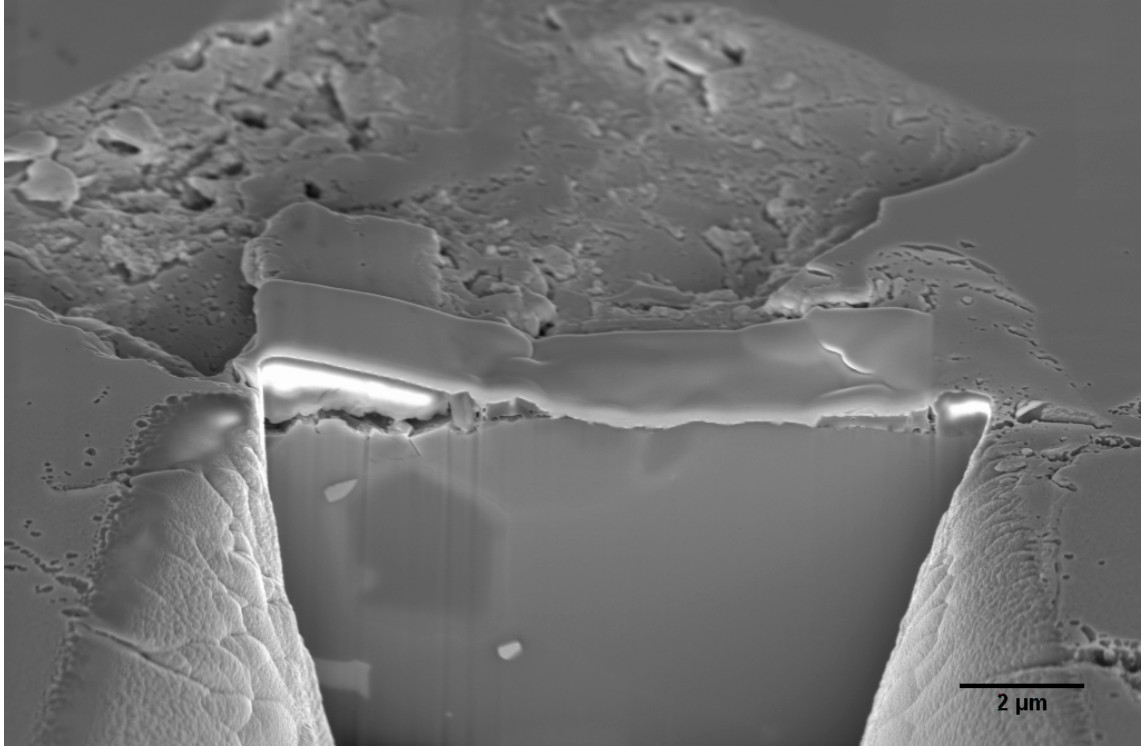

**Figure 6.** FIB cross section of zone 2 (refer to Figure 4b).

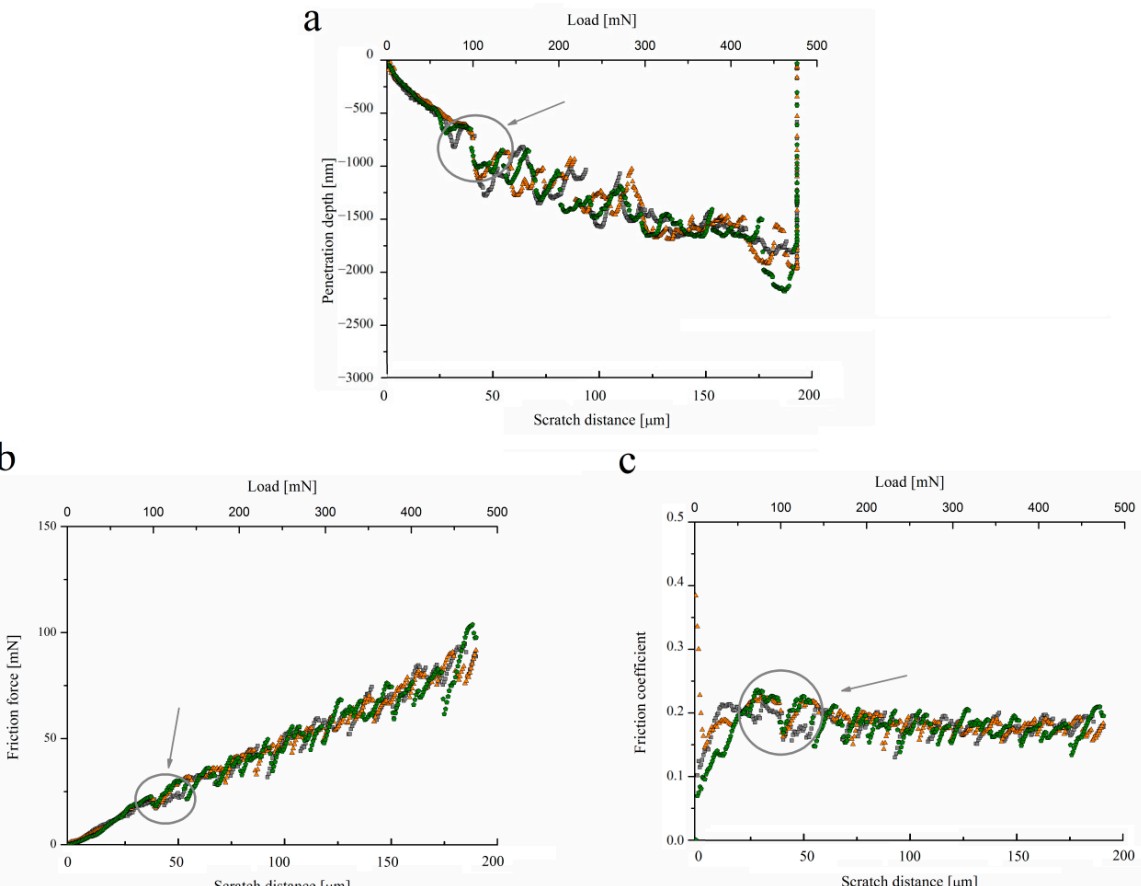

**Figure 7.** Curves of the high-load scratches. Indenter penetration as a function of scratch distance during loading. (**a**) Penetration depth, (**b**) friction coefficient, and (**c**) friction force as a function of scratch distance. Discontinuities recorded in the experiments are marked with circles and signaled by arrows.

The discontinuity discerned in the curves of Figure 7 coincides with the onset of the chipping events evidenced by the damage observed at and below the surface (Figures 4 and 6). Consequently, one can state that for the high-load nanoscratches, the critical load for the film is $P_c \approx 110$ mN. As loads higher than $P_c$ continue to be applied, the discontinuity recorded in both the penetration and friction curves keeps occurring until the maximum load (see Figure 7), which can be related to the repeated chipping observed in the microscopic examination.

It is also worth mentioning that the lateral cracks present within the film, which occur before the chipping and delamination process takes place, were not reflected in the penetration and friction curves recorded.

### 3.3. Damage Micromechanisms

From a general standpoint, fractures are an almost inevitable consequence of highly loaded contacts in the case of ceramic-ceramic coated pairs [32]. When scratching a ceramic coating at relatively low loads (or in the case of thick coatings), fractures are often analogous to those observed in bulk samples of the coating material and damage is contained to the coating. This has been observed in the nanoscratch testing of relatively thick ($t \geq 10$ μm) CVD mullite coatings in a previous investigation by the authors [33]. Conversely, as the scratch load is increased (and/or if the thickness of the coating is reduced), the substrate plays a more prominent role in affecting, or even controlling, fracture behavior.

Furthermore, there is an additional consideration with regard to the analysis of the mechanisms of damage caused by scratching thin coated systems: both normal and lateral

forces (shear stresses) are induced and promoted in the scratched film; thus, different shear-related deformation and failure events may occur within the coated system evaluated.

The sequence of images shown in Figure 8a–d summarizes the damage micromechanisms and their evolution as the scratch load is increased for the coated $3Al_2O_3 \cdot 2SiO_2$/SiC system studied. At low loads (Figure 8a), the scratching indenter only induces radial cracking in the films, and the load is transferred to the substrate in the form of intergranular cracking. This damage event occurs in the load range 0 mN < $P_c$ < 25 mN.

*Scratch damage micromechanisms*

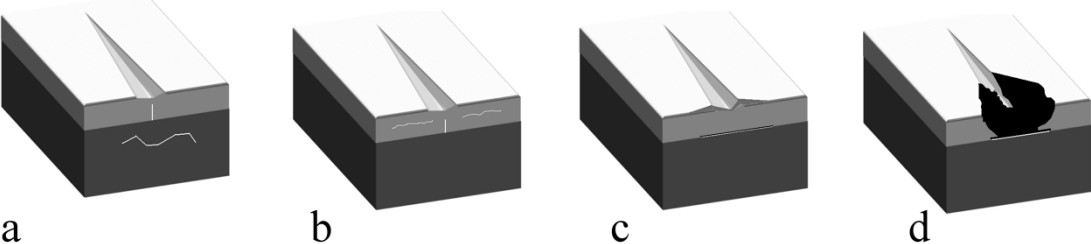

**Figure 8.** Scratch damage micromechanisms under increasing load tests. Radial cracking within the films and intergranular cracking in SiC substrate (**a**), followed by lateral film cracking (**b**) which progresses to film delamination (**c**) and finally film chipping (**d**).

As the load is increased, the scratch causes additional damage events within the film. Lateral cracks occur in addition to radial ones (Figure 8b). Lateral cracks in the film, which precede chipping, can be compared to the ring-type cracks generated when scratching bulk ceramics in conventional scratch tests with conical indenters. This cracking has also been found to be common in brittle coatings deposited on similarly brittle substrates, preceding the delamination of the films in the form of chipping [21,34–36].

It can thus be speculated that the lateral cracks observed are induced in the film by means of shear stresses generated by the lateral movement of the indenter. AFM images of the first 60 μm of one of the high-load scratches are presented in Figure 9a and 9b (height and phase channels respectively). The height contrast obtained in Figure 9a facilitates identification of the critical load for the lateral cracking produced in the mullite film ($P_c \approx 45$ mN), as indicated by arrows in the same figure.

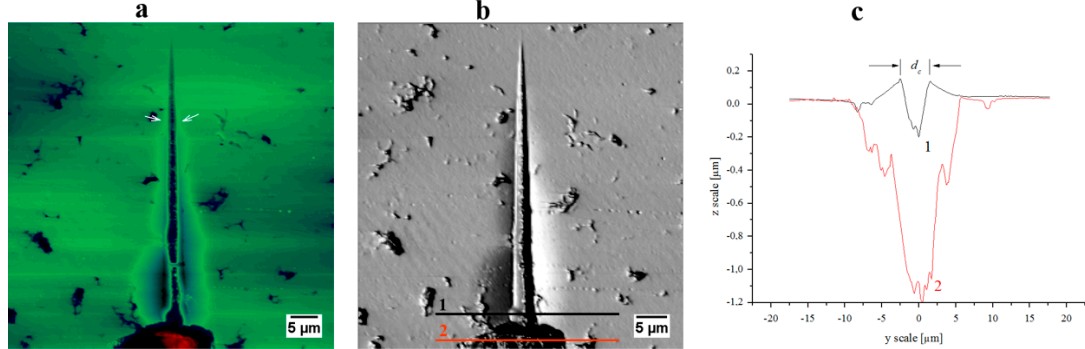

**Figure 9.** AFM images of the first nm of a high-load nanoscratch. (**a**) Height channel and (**b**) phase channel. Onset of lateral cracking in the film is pointed in figure (**a**) with arrows. (**c**) Profile of sample surface right before (1) and after (2) film delamination and chipping. Pore-like features on the top surface of the samples correspond to the irregularities of the mullite coatings after polishing.

As the load increases further, lateral cracks progress through the film until they reach the interface (see Figure 8c). The film then chips out from the substrate, which occurs at a critical load of approximately $P_c \approx 110$ mN (see Figure 8d). At loads higher than $P_c$, any further sliding movement of the scratching tip will result in new cracks progressing until

reaching once again the interface, thus generating new chipping events. This process is repeated, and numerous chippings are evidenced until the maximum load is reached.

### 3.4. Adhesion and Interfacial Toughness

With regard to the nanoscratching of brittle coated systems, it is important to consider that although chipping of the coating in front of the scratching tip is linked to interfacial failure, it may not necessarily be correlated directly with interfacial properties (i.e., adhesion). This would only be the case if cracking in the coating occurred before the event, since the chipping process itself is caused by the kinking of a crack back into and through the coating during delamination [33].

Based on the above, the scenario of delamination should be examined in the studied $3Al_2O_3 \cdot 2SiO_2/SiC$ system. The surface profiles measured just before and after the critical load is reached are displayed in Figure 9c. At the midpoint of the scratch track, it can be seen that the profile measured after chipping coincides with the coating thickness. This indicates that at the moment of chipping, the film is chipped away from the substrate at the interface. This observation, together with the evident film cracking before delamination and chipping, validates the use of the critical load as an effective measure of the structural integrity of the coated $3Al_2O_3 \cdot 2SiO_2/SiC$ system in terms of film adhesion [37–39].

Within the above framework, the models proposed by Laugier [37], Bull et al. [38], Malzbender et al. [40], Thouless [41], and Toonder et al. [30] are implemented here to calculate the adhesion energy ($G_{int}$) of the $3Al_2O_3 \cdot 2SiO_2$ film to the SiC substrate. Equations corresponding to each of the models are included in Table 1 (Equations (1)–(5)), together with the average calculated values of $G_{int}$. An assessment of $G_{int}$ is conducted, accounting for film properties (coefficient of friction between the film and the indenter $\mu_f$; Poisson's ratio $\nu_f$; residual stresses level $\sigma_r$; thickness $t$; elastic modulus $E_f$) as well as scratch-related parameters (critical load for delamination $P_c$; critical stress $\sigma_c$; geometric parameters $L$, $d_c$, and $\beta$, depicted in Figure 4b). Specific parameters used in this investigation are included in Table 2.

**Table 1.** Models for calculating the adhesion energy and corresponding average values of calculated $G_{int}$ and $K_{int.}$

| Model | Equation | Formulation | $G_{int}$ [J·m$^{-2}$] | $K_{int}$ [MPa·m$^{1/2}$] |
|---|---|---|---|---|
| *Laugier* [28] | (1) | $G_{int} = \frac{\sigma_c^2 t}{2E_f}$ | 18.60 | 1.70 |
| *Bull* et al. [29] | (2) | $G_{int} = \frac{P_c^2 t \nu_f^2 \mu_f^2}{2E_f t^2 d_c^2}$ | 5.19 | 0.89 |
| *Malzbender* et al. [30] | (3) | $G_{int} = 0.158 \frac{E_f t^5}{L^4}(\sin\beta)^2 + \frac{t\sigma_r^2}{2}\sqrt{\frac{1}{E_f E_{int}}}$ | 5.45 | 0.92 |
| *Thouless* [31] | (4) | $G_{int} = 0.35 \frac{E_f t^5}{L^4}\left(\frac{\tan\beta + 2a/L}{\tan\beta + a/L}\right)^2$ | 13.81 | 1.46 |
| *Toonder* et al. [25] | (5) | $G_{int} = 1.42 \frac{E_f t^5}{L^4}\left(\frac{a/L+\beta\pi/2}{a/L+\beta\pi}\right)^2 + \frac{t\left(1-\nu_f^2\right)\sigma_r^2}{E_f}$ $+ \frac{3.36(1-\nu)t^3\sigma_r}{L^2}\left(\frac{a/L+\beta\pi/2}{a/L+\beta\pi}\right)$ | 15.05 | 1.52 |

**Table 2.** Parameters used for the calculation of adhesion energy.

| $\mu_f$ * | $\nu_f$ * | $P_c$ [N] | $\sigma_c$ ** [GPa] | $\sigma_r$ * [GPa] | $t$ [µm] | $E_f$ * [GPa] | $\beta$ [rad] | $d_c$ [µm] | $L$ [µm] |
|---|---|---|---|---|---|---|---|---|---|
| 0.25 | 0.22 | 0.11 | 2.12 | 0.04 | 1.2 | 145 | 1.36 | 4.5 | 10 |

* Film parameters extracted from a previous investigation [16]. ** Critical stress ($\sigma_c$) is calculated from the expression provided in the reference [38].

Based on the $G_{int}$ values obtained from the implementation of the models, corresponding values of interface fracture toughness $K_{int}$ can be calculated as follows [39]:

$$K_{int} = \sqrt{\frac{G_{int}E_f}{(1-\nu_f^2)}}$$

(6)

Values of $K_{int}$ determined from Equation (6) are also included in Table 1. A range of adhesion values for $G_{int}$ (5.19–18.60 J·m$^{-2}$) and $K_{int}$ (0.89–1.70 MPa·m$^{1/2}$) are then estimated by means of nanoscratch testing. Despite the relatively large spread of adhesion values found in the literature for different coated systems [37,42], the $G_{int}$ and $K_{int}$ values attained are within the range reported for comparable ceramic coating–substrate pairs [19,24,37,43–45].

Average $G_{int}$ and $K_{int}$ obtained from the different models are graphically presented in the plots in Figure 10a and 10b respectively). For the purpose of comparison, the range of $K_{int}$ and $G_{int}$ values reported in a previous investigation for the same coated system [15], obtained using nanoindentation techniques, are also included in the figure as a gray band on the background of the plots.

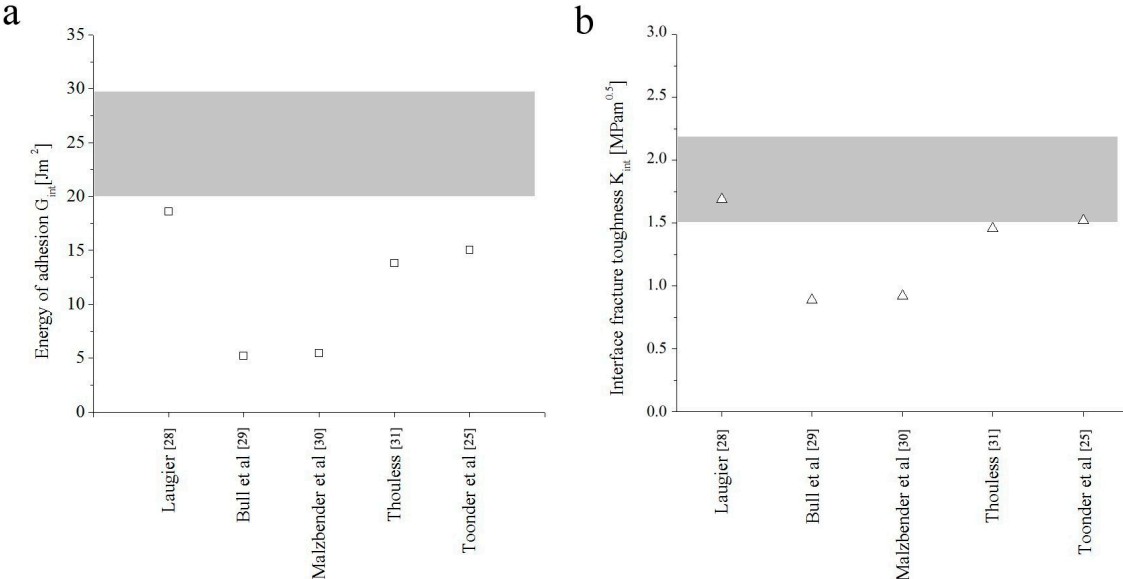

**Figure 10.** (**a**) Adhesion energy and (**b**) interface fracture toughness as calculated from the implemented models. The gray shaded band indicates the domain of the properties previously calculated through nanoindentation [16].

In Figure 10, it can be seen that the adhesion energy and interface fracture toughness values obtained through nanoscratch tests align with those calculated through nanoindentation tests, although the former are slightly lower than the latter. In attempting to rationalize this finding, it should be recalled that in the nanoscratch experiment, in addition to the normal compressive stresses, the indenter tip induces lateral shear stresses and even some tensile stresses that move in line with the indenter tip. Both the shear and tensile stresses applied when performing the scratch tests are greater than those induced by the nanoindentation test, thus provoking early film fracture and delamination at the interface. Comparatively low nanoscratch adhesion values have also been reported for different brittle coated systems with damage features similar to the films studied here [24,44,46].

## 4. Conclusions

The structural integrity of thin $3Al_2O_3 \cdot 2SiO_2$ films of stoichiometric composition deposited on SiC, and that of the $3Al_2O_3 \cdot 2SiO_2$/SiC-coated system intended as a structural unit, were evaluated by means of nanoscratch tests. From the results and corresponding analysis, the following conclusions are drawn:

(1) Nanoscratch testing using increasing applied load introduces controlled and progressive damage in the coated $3Al_2O_3 \cdot 2SiO_2$/SiC system. It was documented that the damage scenario evolves from the early emergence of radial cracks to the appearance of lateral cracks with increasing load; final delamination and repeated chipping take place at higher load values.

(2) Mullite coatings are found to exhibit good adhesion to the silicon carbide substrate. The adhesion energy and interfacial fracture toughness of the coated system under consideration were estimated through critical analysis of the delamination and chipping scenario (induced by nanoscratch testing), combined with the use of different models proposed in the literature. The range of values found for the studied $Al_2O_3 \cdot 2SiO_2$ film to SiC substrate were $G_{int} \approx 5\text{--}20\ \mathrm{J \cdot m^{-2}}$, and $K_f \approx 1\text{--}1.7\ \mathrm{MPa \cdot m^{1/2}}$.

(3) Nanoscratch testing and damage characterization using advanced microscopy techniques are validated here as suitable tools for assessing the structural integrity of these ceramic coating-substrate pairs. Furthermore, they are suggested as potential characterization protocols to monitor changes in the mechanical integrity of these coated systems that may result from exposure to service-like conditions such as thermal loading and thermal fatigue.

**Author Contributions:** Conceptualization, C.A.B., L.L. and E.J.-P.; methodology, C.A.B., L.C. and V.K.S.; formal analysis, C.A.B., L.L. and E.J.-P.; investigation, C.A.B. and E.J.-P.; resources, L.L., E.J.-P. and V.K.S.; writing—original draft preparation, C.A.B., L.L. and E.J.-P.; writing—review and editing, C.A.B., L.L., L.C. and E.J.-P.; funding acquisition, L.L. and V.K.S. All authors have read and agreed to the published version of the manuscript.

**Funding:** This work was partly funded by the Spanish Ministerio de Ciencia, Innovación y Universidades through Grants PID2019-106631GB-C41 and PID2021-126614OB-I00.

**Institutional Review Board Statement:** Not applicable.

**Informed Consent Statement:** Not applicable.

**Data Availability Statement:** Not applicable.

**Conflicts of Interest:** The authors declare no conflict of interest.

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
