# Peer review of "Nanoscratch Testing of 3Al2O3·2SiO2 EBCs: Assessment of Induced Damage and Estimation of Adhesion Strength"

_ceramics, doi:10.3390/ceramics6010040_

Round 1

Reviewer 1 Report

The manuscript researched on the structural integrity of thin 3Al2O3·2SiO2 films of stoichiometric composition deposited on SiC. In this paper the 3Al2O3·2SiO2 coated system intended as a structural unit, were evaluated by means of nanoscratch tests.

Though this work is interesting and contains technical merits for publication, but the English grammar polishing is recommended.

Therefore, after major revision, we recommend publication of the paper.

For example

1.      There are wrong with the numbering between headings

l  Introduction

2.      The 16rd line

The “are” should be “is”.

3.      The 19rd line

The “rised” should be “raised”.

4.      As we all know, environmental barrier coatings are mainly used in high temperature service environments. Does it meet the performance requirements when tested at room temperature?

l  Introduction

5.      The 31rd line

The “achieving” should be “achieve”.

6.      The 35rd line

The “is” should be “are”.

7.      The 76rd line

The “were” should be “was”.

l  Materials and Methods

8.      The 98rd line

The “grey” should be “gray”.

9.      The FIB's slice shows only a partial picture. Is the overall picture consistent as described in the article?

10.  The 140rd line

The “at” should be “on”.

l  Results and Discussion

11.  Only the friction experiment of words is carried out in this paper. Are the observed phenomena repeatable?

12.  The 182rd line

The “coated” should be “coating”.

13.  The 183rd line

The “portion” should be plural.

14.  The 247rd line

The morphology of “nature” may be wrong, please check.

15.  In this paper, the maximum load and the scratch length are changed at the same time. If only one condition is changed, is the phenomenon consistent with the experiment?

16.  What is the basis for the selection of maximum load and scratch length in the experiment?

17.  In the figure 7.b and 7.c

Why does the transverse crack inside the scale not appear in figure 7 b and c?

l  Conclusions

18.  The 416rd line

The “for” should be “of”.

l  Reference

19.  The format of some reference are inconsistent, and you should unify the format.

Reviewer 2 Report

The manuscript is intended to explore the structural integrity intrinsic of the mullite films on the SiC substrates by increasing load nanoscratch tests. The configuration and morphology of the damage induced by the nanoscratch tests and the related micromechanisms were elaborated. In the current submission, however, major editorial revision and further supplement are needed.

1. There are many English mistakes to be listed one by one.

2. Prior to deposition, the SiC substrates are needed to be sandblasted?

3. The thickness of the mullite film has a significant influence on the damage morphology. It is important to indicate the number of specimens of each load tested and the standard deviation of the mullite thickness.

4. Why the mullite film exhibited the considerable elastic recovery (difference in penetration) after the scratch test?

5. Labeling the substrate and coating in Fig.3 is helpful to understand the described content.

6. "As the scratch load is raised (and/or if the thickness of the coating is reduced), the substrate plays an more protagonist role in affecting, or even controlling, fracture behavior". In this work, whether the substrates affect the related damage behavior?

Reviewer 3 Report

The article is devoted to the study of the mechanical properties of films (3Al2O3•2SiO2) obtained by chemical vapor deposition. This line of research is quite interesting, since the proposed film compositions have great prospects for practical application in a wide range of areas. At the same time, studies aimed at determining the resistance of these films to external mechanical influences are of high importance, since they allow answering a number of questions that determine the area of possible application of these structures in practice. The authors analyzed a large amount of experimental data, and the presented dependences have a logical construction and a completed study. In general, the article can be accepted for publication after the authors answer a number of questions from the reviewer that arose during the analysis of the article and its consideration.

1. In the abstract, the authors should give some details about the object of study (3Al2O3•2SiO2) thin films, since these structures are quite well known, and the method proposed by the authors is very promising for these films.

2. Figure 1 shows an image of a side cleavage of the objects under study; it is recommended that the authors give the characteristic dimensions of the layer thickness, which will make it possible to determine their characterization.

3. According to the presented image data of the samples after testing, the presence of small pores and microcracks is observed in the layer structure. The authors should give an explanation of the nature of their origin.

4. The proposed model of cracking requires additional explanations, in particular, what is the reason for the increase in cracking and chipping of samples under high loads.

5. Authors are also encouraged to provide any additional data on the characterization of the original samples.

Round 2

Reviewer 3 Report

The authors answered all the questions, the article can be accepted for publication.